

# Power Plant Fuel Switching and Air Quality in a Tropical Forested Environment

Adan S. S. Medeiros (1,2), Gisele Calderaro (1,2), Patricia C. Guimarães (1,2), Mateus R.

Magalhaes (1,2), Marcos V. B. Morais (3), Sameh A. A. Rafee (3), Igor O. Ribeiro (1,2), Rita V.

Andreoli (1), Jorge A. Martins (3), Leila D. Martins (3), Scot T. Martin[*] (4), Rodrigo A. F.

Souza[*] (1)

(1) Amazonas State University, Manaus, Amazonas, Brazil

(2) National Institute of Amazonian Research, Manaus, Amazonas, Brazil

(3) Federal University of Technology, Paraná, Av. Dos Pioneiros, 3131, Londrina, 86047-125, Brazil

(4) Harvard University, Cambridge, Massachusetts, USA

E-mail: *scot_martin@harvard.edu, souzaraf@gmail.com*

*Atmospheric Chemistry and Physics*

[*]To Whom Correspondence Should be addressed





**Abstract**

How a changing energy matrix for power production affects air quality is considered for

an urban region in a tropical, forested environment. Manaus, the largest city in the central

Amazon basin of Brazil, is in the process of changing its fossil fuel power energy matrix from

entirely fuel oil and diesel to nearly entirely natural gas across an approximately ten-year period.

Three scenarios of urban air quality, specifically afternoon ozone concentrations, were simulated

using the Weather Research and Forecasting (WRF-Chem) model. The first scenario used fuel

oil and diesel for power production, which was the reality in 2008. The second scenario was

based on the fuel mix from 2014, the most current year for which data were available. The third

scenario considered nearly complete use of natural gas for power production, which is the

anticipated future, possibly for 2018. For each case, inventories of anthropogenic emissions were

based on power generation, refining operations, and transportation. Transportation and refinery

operations were held constant across the three scenarios to focus on effects of power plant fuel

switching in a tropical context. The results of the simulations indicate that a change to natural

gas significantly decreases maximum afternoon ozone concentrations over the population center,

reaching reductions of 73% (110 to 30 ppb) on the most polluted days. $NO_x$ and CO emissions

decreased by approximately 89% and 55%, respectively, after the complete change in the energy

matrix. The sensitivity of ozone concentrations to the fuel switchover is consistent with a $NO_x$-

limited regime, as expected for a tropical forest having high emissions of biogenic volatile

organic compounds, high water vapor concentrations, and abundant solar radiation. Thus,

policies favoring the burning of natural gas in place of fuel oil and diesel have great potential for

ozone reduction and improve air quality for growing urban regions located in tropical, forested

environments around the world.





## 1. Introduction

The evolution of modern civilization is closely associated with obtaining and distributing energy at large scale (Price, 1995). Although electrical power production for Brazil as a whole is obtained mostly by hydroelectric plants (ANEEL, 2008), in today's Amazon region, constituting the largest tropical forest in the world (Behling et al., 2001), electricity is provided largely by thermal power generation that burns fossil fuels (ELETROBRAS, 2014a). The Amazon region is of vital importance for the functioning of both regional ecosystems and climate (Fisch et al., 1998;Nobre et al., 2016). Topics for research in recent years have included the relationship between the biosphere and the atmosphere in the Amazon (Fan et al., 1990;Stark et al., 2015), the impacts of land use change (Dickinson and Kennedy, 1992;Fearnside, 2003;Paula et al., 2014;Wertz-Kanounnikoff et al., 2016), and the consequences of urbanization, population growth, and increased anthropogenic emissions to the composition of the atmosphere (Shukla et al., 1990;Potter et al., 2001;Wright, 2005;Malhi et al., 2008;Martin et al., 2016).

The population of northern Brazil has grown exponentially in recent decades. In the last 50 years (1960-2010), the urban population of the region increased from about 1 to 11 million, while the urban population of Brazil grew from 32 to 160 million in the same period (IBGE, 2010). This growth in the northern region is linked to public politics to expand the occupation, exemplified by the establishment in 1967 of a free trade zone in Manaus in central Amazonia. In 2014, this concession was extended until 2073 (Queiroz, 2014), indicating a continuation in the rate of population growth for the region in the coming decades. Continued growth can be expected of the power generating systems to supply the population and industry.

In this context is Manaus, Amazonas, the financial, corporate, and economic center of northern Brazil. It has a population of two million and is the seventh largest city in Brazil (IBGE,



2015). The population in recent time has increased nearly every year by 50,000 persons due to

internal migration motivated by the large industrial district, an area that receives tax exemption

from the government. Population growth continually increases the demand for land, energy, and

power, leading to the loss of adjacent forest and the degradation of air quality in the region

(Cropper and Griffiths, 1994).

In 2009, a 650-km natural gas pipeline was inaugurated, linking a production region in

Urucu, Amazonas, to Manaus (Soares et al., 2014). From an operational and cost point of view,

an uninterrupted fuel supply and the direct distribution from the source to the end user, such as

provided by the natural gas pipeline, significantly reduce both costs and related emissions for the

transport of fuels on trucks and ships (Neiva and Gama, 2010). With the supply of natural gas,

the power plants of Manaus have been adjusting to the economic conditions of the changed fuel

mix, replacing fuel oil and diesel with natural gas across approximately a ten-year period.

Although the historical change was not motivated at the policy level by environmental drivers,

the change nonetheless represents a unique opportunity to evaluate how fuel switching can affect

air quality, especially in regard to little-studied tropical forest environments.

Emission factors of pollutants and pollutant precursors differ greatly between fuel oil and

diesel on the one hand and natural gas on the other, and these emissions affect regional air

quality and human health (Vitousek et al., 1997;Holgate et al., 1999). Air pollution can lead to

arterial vasoconstriction (Brook et al., 2002), cytogenetic damage in lymphocytes (Holland et al.,

2015), and chronic obstructive pulmonary disease (COPD) (Schikowski et al., 2014), and asthma

immunopathogenesis (Alexis and Carlsten, 2014). The World Health Organization (WHO)

provides recommendations on the thresholds of pollutant concentrations, such as ozone,



particulate matter, nitrogen dioxide, and sulfuric dioxide, above which human health is adversely

affected (WHO, 2006).

Ozone is the criteria air quality pollutant considered herein. The interactions among

oxides of nitrogen ($NO_x$), volatile organic compounds (VOCs), water vapor, and sunlight

combine to produce ozone (Seinfeld and Pandis, 2006). The ratio of $NO_x$ to VOC concentrations

is of fundamental importance for the production rate of ozone. In tropical, forested Amazonia,

biogenic volatile organic compounds (BVOCs) are emitted from the forest and are naturally

abundant (Fehsenfeld et al., 1992;Kesselmeier and Staudt, 1999;Karl et al., 2007;Jardine et al.,

2015;Jokinen et al., 2015;Yáñez-Serrano et al., 2015;Liu et al., 2016). By comparison, human

activities can significantly elevate $NO_x$ concentrations above background concentrations (Delmas

et al., 1997;Lamarque et al., 2010;Daskalakis et al., 2016). For this reason, economic activities

and policy decisions that can affect anthropogenic $NO_x$ emissions deserve special attention in the

context of Amazonia. A quantitative understanding of how an anthropogenically perturbed

VOC:$NO_x$ ratio affects ozone production in this region is, however, not trivial. Compared to

temperate urban regions that have been studied in greater detail for ozone production, the

tropical region has more intense solar radiation and higher water vapor concentrations (Kuhn et

al., 2010). Regional modeling is an important approach for understanding the linked effects

(Potter et al., 2001;Isaksen et al., 2009).

Three different cases are considered herein to investigate how an altered energy matrix

for power production affects ozone concentrations in the urban area of Manaus. Power

generation is the major source of pollutant emissions to the region, rather than mobile sources, as

discussed by Rafee et al. (2017). For Case A, fuel oil and diesel are used for power production,

which was the reality in 2008. The Urucu pipeline began initial, albeit small, shipments of





natural gas in 2010, with increasing amounts every year thereafter. By 2014, natural gas had

increased from 0% to 65% of the energy matrix for power production. Case B corresponds to the

energy matrix of 2014. Case C considers the nearly complete use of natural gas for power

production, which is the planned future, possibly for 2018.

**2. Model Description**

**2.1 WRF-Chem**

Simulations were carried out using the Weather Research and Forecasting model fully

coupled to a chemical module (WRF-Chem version 3.6.1) (Grell et al., 2005). The WRF

configuration included the treatment of Lin et al. (1983) for cloud microphysics, MM5 for

surface layer (Grell et al., 1994), Noah for land surface (Chen et al., 1997), Yonsei University for

boundary layer (Hong et al., 2006), Goddard for short-wave radiation (Chou and Suarez, 1999),

rapid radiative transfer model for long-wave radiation (Mlawer et al., 1997), and Grell and

Freitas (2013) for cumulus clouds. The modelling approach with these parametrizations have

been studied (Ying et al., 2009;Misenis and Zhang, 2010;Gupta and Mohan, 2015), showing

sensitivity to capture the effects of changing emissions inventory.

Two nested domains were employed (Figure 1). An outer domain (1) had a resolution of

10 km and a dimension of 1050 km × 800 km. This domain employed reanalysis data from

Climate Forecast System Reanalysis (CFSv2). An inner domain (2) had a resolution of 2 km and

a dimension of 302 km × 232 km. This domain included the study area. The domain had

boundary conditions based on interpolation of domain 1. The grid center was the same for both

domains (2.908° S and 60.319° W). The model spin-up time was 24 h, followed by 72 h of

simulation. In this way, ten simulations were carried out for a one-month period. This approach

balanced between computational time and numerical diffusion.





Meteorological fields used in this work were obtained from reanalysis data of the

National Center for Environmental Prediction (NCEP) at a spatial resolution of 0.5° × 0.5° and a

time resolution of 6 h from February 1 to 28, 2014. NCEP data are based on the Climate Forecast

System Reanalysis (Saha et al., 2011). Land cover was based on data from the Moderate

Resolution Imaging Spectroradiometer (MODIS) (Rafee et al., 2015). For this region, the

climatogical rainfall in February is 290 mm, which can be compared to a maximum of 335 mm

in March and a minimum of 47 mm in August (Ramos et al., 2009). During this period

contributions by biomass burning were most often negligible (Martin et al., 2016).

For the chemical part of the model, anthropogenic and biogenic emissions of gases were

considered (described in section 2.2). The Regional Acid Deposition Model Version 2 (RADM2)

was used as chemical mechanism (Chang, 1991). Initial and boundary conditions from trace

gases were obtained from the Model for Ozone and Related Tracers (MOZART-4) (Emmons et

al., 2010).

**2.2 Emissions**

Forest, vehicle, power plant, and refinery emissions were considered. Biogenic emissions

were based on MEGAN (Model of Emissions of Gases and Aerosols from Nature, version 2.1)

(Guenther et al., 2012). MEGAN gathers a wide range of information about the global

distribution of emissions from vegetation and soil. It considers about 150 different compounds

following the approximate distribution: 50% of isoprene, 30% of methanol, ethanol,

acetaldehyde, acetone, α-pinene, β-pinene, t-β-ocimene, limonene, ethane, and propene, 17% of

another twenty compounds (mostly terpenoids), and of 3% among another 100 compounds.

Variability in emissions takes into account the type of vegetation, the seasonality based on

temperature and leaf area index (LAI), the intensity of incident light, and the soil moisture.





For vehicle emissions, a vehicle count for Manaus of 600,000 was considered

(DENATRAN, 2014). The breakdown of vehicle by type, daily travel distances, and emission

factors is listed in Table 1 (ANP, 2014). The average age of the Manaus fleet is 5 years. The

methodology of Martins et al. (2010) was used to distribute the emissions spatially based on

nighttime light intensity observations of the Defense Meteorological Satellite Program –

Operational Linescan System (DMSP-OLS). These observations were assumed to correlate with

overall daily patterns of vehicle traffic, and more details about the methodology to obtain the

vehicular emissions are described in (Andrade et al., 2015).

For stationary sources related to power production, a survey of the locations of power

plants in Manaus region was conducted. Although the city of Manaus has a large industrial park,

these industries mostly produce electronic products and burn little fuel directly. Instead, power

plants and a single large refinery are major emitters. The data of installed and generated power

and the fuel used were obtained for each plant in Manaus urban zone (Table 2). The locations of

these power plants are shown in Figure 1. The emission factors for power plants were based on

the USA Environmental Protection Agency (EPA) using the median value of the emission

factors for power generation by the different fuel types (Table 3). The median consumption of

fuels and the average power in 2014 are also listed in Table 3 (ELETROBRAS, 2014b). Another

major source of pollution in the region is the refinery Isaac Sabbá, with the capacity to process

$7.3 \times 10^6$ liters of oil per day (PETROBRAS, 2016). The emission factors of the refinery are

listed in Table 3 (DeLuchi, 1993). The speciation of the VOC emissions for both power plants

were performed based also on the EPA emission for each type of fuel.





**2.3 Scenarios**

Simulations were performed to evaluate ozone concentrations for three different

scenarios. The first scenario (Case A) was based on emissions of historical Manaus before the

gradual process of fuel switching began in 2010. It corresponded to an energy matrix of 100% oil

or diesel for power generation. Because a gradual change in the energy matrix took place, the

second scenario (Case B) considered the mix of oil, diesel, and natural gas used in 2014 for

power generation. In 2014, 65% of the power was generated by natural gas and the remaining

35% by oil or diesel (ELETROBRAS, 2014b). The third scenario (Case C) used an energy

matrix of 100% natural gas, removing all oil and diesel from power production in Manaus. This

scenario represents the anticipated situation for the Manaus region within the next several years.

Table 2 lists the fuel mix of each case. All three scenarios also include a baseline contribution of

24% by regional hydropower.

In order to compare only the effects of the change in the type of fuel used, the same

matrix of power plants was used for the three scenarios. Although the combined capacity of

power production increased in recent years following the population and energy demand growth,

this change was omitted so the comparative study of the effects of fuel type on air quality could

be isolated. Likewise, vehicle emissions were on purpose held constant for the three fuel

scenarios considered herein to focus on effects of power plant fuel switching in a tropical

context. In this regard, the intent of the analysis herein was to represent the effects of power

plant fuel switching on air quality in a tropical forested environment in a general yet realistic

sense by selection of a representative urban environment. The intent was not an actual simulation

of the city of Manaus, which would necessitate adjustment of transportation, industry, power,

land use, and other aspects of urban growth corresponding to the year of each case. The study





herein was also restricted to the wet season, again to focus on shifts in the energy matrix and

avoid the complicating effects of biomass burning prevalent in the dry season.

Table 4 show the daily emissions of CO and $NO_x$ per group of emissions. The vehicles

and refinery emissions are the same in all three cases. Together, the CO emissions corresponds to

16%, 22%, and 36% for Cases A, B, and C, respectively. For $NO_x$, they are 2% for Case A, 5%

for Case B, and 18% for Case C. Thus, it can be concluded that the emissions of ozone

precursors by vehicles and refinery are the smallest component of anthropogenic emissions in

Manaus, even after the fuel change in energy matrix. From Cases A to B, total CO and $NO_x$

emissions has decreased by 25% and 60%, respectively, while the entire change of fuel use on

power plants (Case A to C) reduces emissions over 55% and 89%, respectively.

**3. Results and Discussion**

Figure 2 shows a box-whisker plot for all days and afternoon times of the simulations for

each case. The time period of 12:00 to 16:00 (local time) was selected for analysis because it

represents the maximum ozone concentration, which is fundamentally linked to photochemistry.

For the statistical analysis of Figure 2, an area of 10 km × 10 km centered on Manuas was taken

to assess ozone concentrations in the populated urban area. The black box in Figure 3 represents

this region. The analysis represented in Figure 2 shows that there was strong variation of ozone

concentrations during the simulated month.

The inter-case variability in ozone concentration across Cases A, B, and C in Figure 2

arose from differences in the energy matrix for power production. A partial shift from diesel and

oil to natural gas (i.e., Cases A and B) did not greatly shift ozone concentrations, on either

polluted or clean days. However, a complete shift to natural gas (i.e., Case C) considerably

reduced ozone concentrations in the urban region. Maximum afternoon ozone concentrations



decreased by 73% (110 to 30 ppb) for the three most polluted days of the simulated month. The

intra-case variability in ozone concentration for different days was tied to meteorological

differences across the month. Some days were sunny, favoring the photochemical process of

ozone formation, whereas other days were overcast or rainy. On poor weather days, the

additional pollution from Manaus contributed to small or negligible additional ozone production.

Figure 3 shows examples of the spatial distributions associated with each of Cases A, B,

and C. Each panel shows a map of the afternoon ozone plume for the afternoon of February 1,

2014. Maps of afternoon means and standard deviations across the full simulation period are

shown in Figure S1 of the Supplement. The ozone plume spreads downwind Manaus, in

agreement with observations reported by Kuhn et al. (2010) and Martin et al. (2016), showing

that the pollution produced in Manaus urban region affects regional air quality, and reaches other

cities downwind Manaus, like Careiro, Iranduba, and Manacapuru. The qualitative spatial pattern

of the ozone plume is similar among the cases, as explained by the use of identical meteorology,

where there is a predominance of easterlies. The concentrations, however, have strong

differences. From case A to B, the concentrations inside the plume are close, in agreement with

the results for Figure 2. For Case C, there is a major decrease in ozone footprint not only in

Manaus, but especially in the cities downwind Manaus, reinforcing the importance of the fuel

switch.

Figure 4 presents a difference analysis between historical practice (i.e., Case A) and

future plans (i.e., Case C) to finalize the foregoing points related to Figures 2 and 3. This

difference represents a shift in the entire energy matrix for power production from oil and diesel

to that of natural gas. The left panel of Figure 4 shows the difference map for a single day

corresponding to the plots of Figure 3. Ozone concentrations decrease by approximately 50 ppb





in the center of the plume. The right panel shows a box-whisker plot of difference values in the

afternoon period across the month, corresponding to the plots of Figure 2. Days 1, 5, 8, 12, 15,

and 16 had the highest differences between the two scenarios, indicating that these days were the

sunniest and most polluted. For the other days of the month, the median difference was very

close to zero, indicating that these days were overcast or had high levels of convection that

brought in clean air.

For comparative studies, Mena-Carrasco et al. (2012) studied the benefits of using natural

gas instead of diesel with respect to air quality and human health. The study carried out in for

Santiago, located in the central region of Chile with Mediterranean climate, showed that the use

of natural gas instead of diesel in urban buses could reduce drastically the emissions and

concentrations of particulate matter. Krotkov et al. (2016) reported changes in $SO_2$ and $NO_2$

concentrations in the last ten years using Ozone Monitoring Instruments (OMI) satellite data.

Stands out, similarly what has been happening in Manaus, a 40% decrease in eastern US $NO_2$

concentrations over the last decade due to emissions regulation and technological improvements.

On the other hand, India's $NO_2$ concentrations increased 50% from the growth of coal power

plants and industrial activities. In this context, even though anthropogenic emissions in Amazon

region are low compared to other regions of the world, such as Mexico City (Molina et al.,

2010), São Paulo (Silva Junior and Andrade, 2013), and Los Angeles (Haagen-Smit, 1952), the

study results of Figure 4 demonstrate the significant sensitivity of Amazonia to anthropogenic

emissions. The relationship between the Manaus emissions and the vast biogenic emissions

constitutes an important scenario to study the atmospheric chemistry feedbacks.

In summary, the results show that the altered energy matrix significantly influences air

quality, as gauged by the maximum afternoon ozone concentration. The large diffe7rences



between Cases A and C show that the burning of fuel oil and diesel have enormous potential for

regional ozone production. Conversely, substitution with natural gas has an excellent effect for

comparative air quality and human health. The results also emphasize the high sensitivity over

this tropical forest to even small amounts of pollution, as amplified by the high solar irradiance

and water vapor concentrations in an environment of plentiful BVOC emissions. Specifically, the

significant decrease in $NO_x$ emissions from Case A to B resulted in no strong differences in

ozone concentrations whereas, conversely, the smaller increase from Case C to B resulted in

large ozone production. This nonlinear behavior of ozone concentration with respect to pollution

is linked to the chemical cycles of the ozone production, most specifically related to the $NO_x$

limitation or not (Lin et al., 1988). Frost et al. (2006) likewise showed that decreases in $NO_x$

emissions from power plants in the eastern USA have significantly affected regional ozone

concentrations. The results herein suggest that the anticipated coming complete conversion to

natural gas for power production should significantly reduce ozone concentration in the Manaus

urban region, even as smaller municipalities throughout the Amazon basin (two-thirds the size of

the continental USA) continue to burn diesel for power production. Altering the energy matrix in

this regions is dependent on continued development of infrastructure for use of natural gas or

making connections to the national grid and continued developments in the use of hydropower

(Domingues, 2003;Tundisi, 2007;ANEEL, 2008)

**Acknowledgments**

We acknowledge support from the Central Office of the Large Scale Biosphere Atmosphere

Experiment in Amazonia (LBA), the National Institute of Amazonian Research (INPA), the

Amazonas State University (UEA), the Financier of Studies and Projects (FINEP), and the



Atmospheric System Research (ASR) program of the Office of Biological and Environmental Research, Office of Science, United States Department of Energy (DOE). The first author thanks the Brazilian Coordination for the Improvement of Higher Education Personnel (CAPES) for the grant scholarship, linked to the doctoral program in Climate and Environment (CLIAMB). The work was conducted under 001030/2012-4 and 400063/2014-0 of the Brazilian National Council for Scientific and Technological Development (CNPq), and 062.00568/2014 of the Amazonas State Research Foundation (FAPEAM), which is also acknowledged by the correspondent authors for the Senior Visitor Research Grant.

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





**List of Figures**





**Figure 3.** Maps of near-surface ozone concentrations for historic emissions (Case A), present-day emissions (Case B), and planned future emissions (Case C). Ozone concentrations correspond to 15:00 (local time) near the surface for the meteorology of February 1, 2014. The local river system is shown in the background. The black box, centered over the population center of Manaus, represents the averaging area used in the calculations for Figure 3.

**Figure 4.** Difference analysis for historic compared to planned future emissions (i.e., Case A minus Case C). (left) Difference map of ozone concentrations (cf. Figure 3). (right) Box-whisker plot of differences in afternoon ozone concentrations during the one-month time series (cf. Figure 2).



**Table 1.** Manaus transportation fleet. Percent composition, daily travel distance, and emission factors are listed for different vehicle types in 2014 (ANP, 2014 ).

| Type (fuel) | % | Daily travel distance (km) | CO (g km$^{-1}$) | NO$_x$ (g km$^{-1}$) |
|---|---|---|---|---|
| Light vehicles (gasoline) | 21.6 | 48.2 | 5.43 | 0.34 |
| Light vehicles (ethanol) | 2.5 | 48.2 | 12.0 | 1.12 |
| Light vehicles (flex) | 42.1 | 48.2 | 5.13 | 0.32 |
| Urban bus (diesel) | 1.9 | 208.3 | 4.95 | 9.81 |
| Trucks (diesel) | 3.2 | 304.7 | 4.95 | 9.81 |
| Pickup trucks (diesel) | 3.9 | 49.9 | 4.95 | 9.81 |
| Motorcycles (gasoline) | 24.8 | 27.9 | 9.15 | 0.13 |


**Table 2.** Power plant information in the study region in 2014. Numbered pins in Figure 1 correspond to the power plants listed here. Plant name, the fuel mix for Cases A, B, and C, the power generated (annual average), the power capacity, and the plant location are listed. Fuel types are abbreviated as fuel oil (F), diesel (D), and natural Gas (G) (ELETROBRAS, 2014b). In addition to these plants, there is a hydroelectric power plant (Balbina) 140 km north of Manaus {1.91° S, 59.57° W}, with a power generated of 250 MW and a power capacity of 250 MW (Fearnside, 2005). In 2013, Manaus became linked to the Brazilian national grid (ANEEL, 2013).

| | Power Plant | Case A | Case B | Case C | Power Generated (MW) | Power Capacity (MW) | Capacity Factor | Location |
|---|---|---|---|---|---|---|---|---|
| 1 | Aparecida | D | D (5%) / G (95%) | G | 135.2 | 200.0 | 0.68 | {3.13° S, 60.03° W} |
| 2 | Mauá / Electron | F / D | F (7.8%) / G (35.6%) / D (56.6%) | G | 255.3 | 728.9 | 0.35 | {3.12° S, 59.93° W} |
| 3 | Flores | D | D | G | 41.4 | 94.6 | 0.44 | {3.07° S, 60.02° W} |
| 4 | Cidade Nova | D | D | G | 8.6 | 22.8 | 0.38 | {3.03° S, 59.97° W} |
| 5 | São José | D | D | G | 18.0 | 60.9 | 0.30 | {3.06° S, 59.95° W} |
| 6 | Iranduba | | D | G | 21.9 | 54.7 | 0.40 | {3.20° S, 60.17° W} |
| 7 | Breitener Tambaqui | F | G | G | 61.9 | 60.0 | 1.03 | {3.11° S, 59.94° W} |
| 8 | Breitener Jaraqui | F | G | G | 61.5 | 60.0 | 1.03 | {2.99° S, 60.03° W} |
| 9 | Ponta Negra | F | G | G | 65.1 | 60.0 | 1.09 | {3.09° S, 60.07° W} |
| 10 | Manauara | F | F (10.9%) / G (89.1%) | G | 64.3 | 60.0 | 1.07 | {2.95° S, 60.02° W} |
| 11 | Cristiano Rocha | F | F (10.5%) / G (89.5%) | G | 68.0 | 65.0 | 1.05 | {2.89° S, 60.03° W} |
| | Refinery | | | | | | | {3.14° S, 59.96° W} |
| | Total | | | | 801.2 | 1,466.9 | | |





**Table 3.** Emission factors for consumption of fuel oil, diesel, and natural gas in power production (median values obtained from EPA (1998)). The fuel consumption factor for power production is also listed (ELETROBRAS, 2014b). The emission factors for oil refining are shown in the right column (DeLuchi, 1993).

| | Fuel oil (g $L^{-1}$) | Diesel (g $L^{-1}$) | Natural gas (g $m^{-3}$) | Refinery (g $L^{-1}$) |
|---|---|---|---|---|
| CO | 0.60 | 3.65 | 0.97 | 0.45 |
| $NO_x$ | 3.90 | 36.20 | 2.50 | 0.56 |
| Fuel Consumption | 0.29 (L $kWh^{-1}$) | 0.38 (L $kWh^{-1}$) | 0.25 ($m^3$ $kWh^{-1}$) | |



**Table 4.** Emissions of carbon monoxide (CO) and nitrogen oxide ($NO_x$) by vehicles, power plants, refinery, and total for historic emissions (Case A), present-day emissions (Case B), and planned future emissions (Case C).

| | Case A (kg day$^{-1}$) | Case B (kg day$^{-1}$) | Case C (kg day$^{-1}$) |
|---|---|---|---|
| CO (vehicles) | 804 | 804 | 804 |
| CO (diesel power plants) | 11,647 | 6,126 | 0 |
| CO (fuel oil power plants) | 1,755 | 186 | 0 |
| CO (natural gas power plants) | 0 | 3,080 | 4,692 |
| CO (refinery) | 1,825 | 1,825 | 1,825 |
| **Total CO** | **16,032** | **12,022** | **7,321** |
| | | | |
| $NO_x$ (vehicles) | 435 | 435 | 435 |
| $NO_x$ (diesel power plants) | 115,520 | 40,760 | 0 |
| $NO_x$ (fuel oil power plants) | 11,410 | 1,212 | 0 |
| $NO_x$ (natural gas power plants) | 0 | 7,889 | 12,015 |
| $NO_x$ (refinery) | 2,117 | 2,117 | 2,117 |
| **Total $NO_x$** | **129,482** | **52,414** | **14,567** |



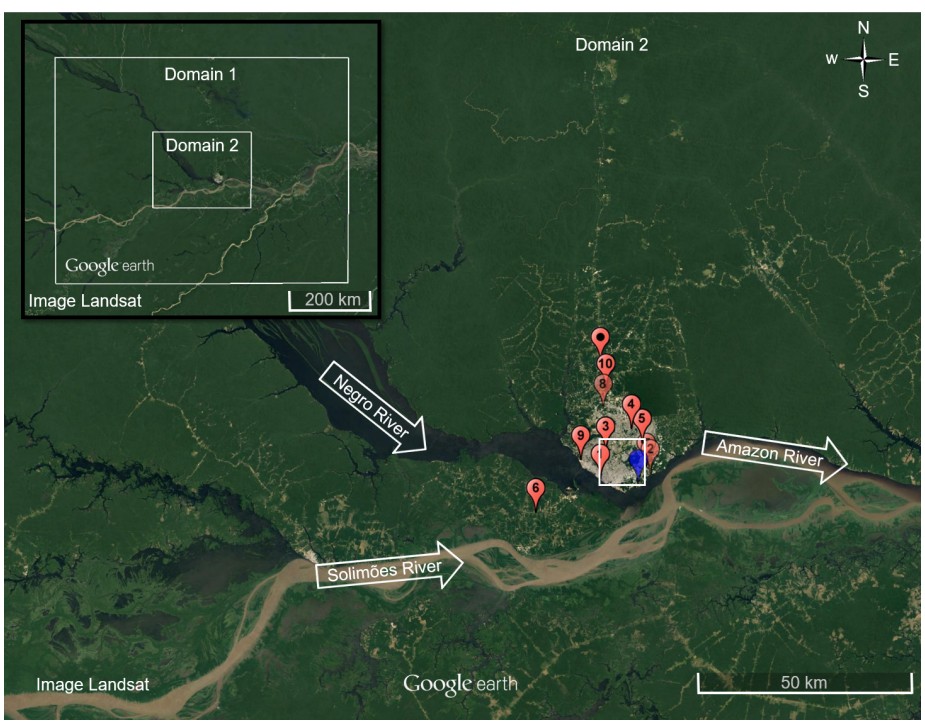

**Figure 1**





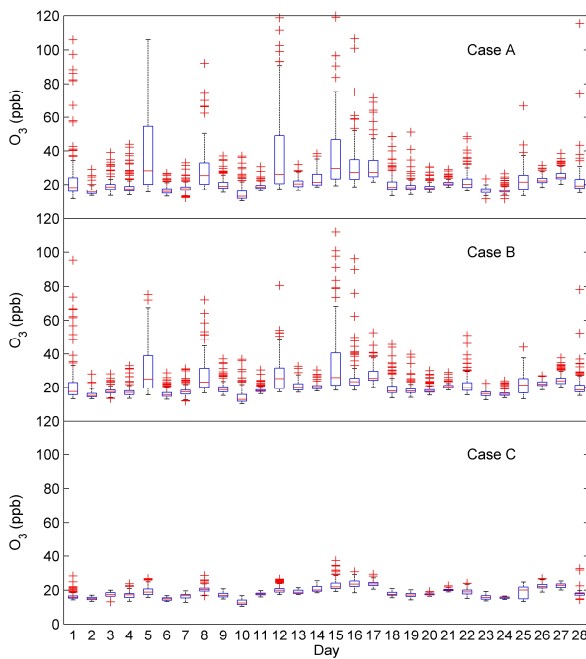

**Figure 2**





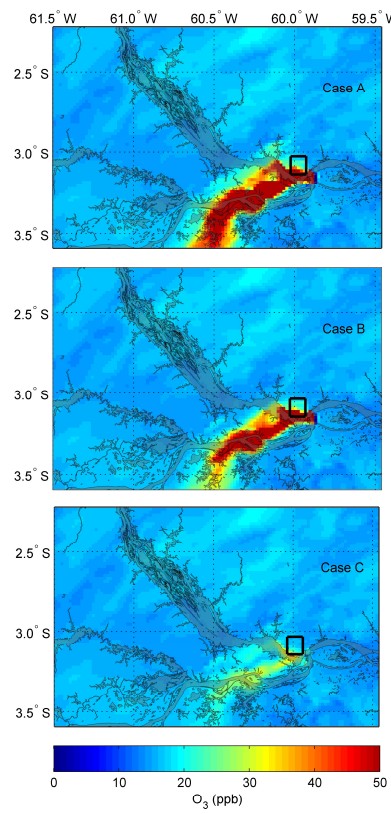

**Figure 3**



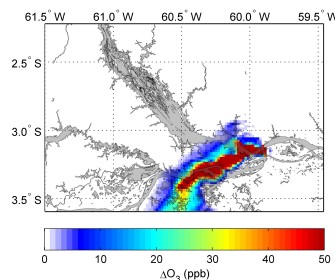

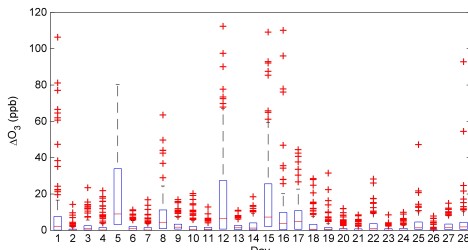

**Figure 4**