# Peer review of "Power Plant Fuel Switching and Air Quality in a Tropical Forested Environment"

_Atmospheric Chemistry and Physics, 2016_

## Short Comment (SC1) · 11 Mar 2017

The manuscript brings an interesting contribution to the evaluation of the air pollution conditions associated with the production of thermoelectric energy in Manaus, Amazonas, Brazil. In particular, it presents modelling evidence based on an optimistic scenario of reducing the levels of atmospheric pollutants that participate in photochemical reactions to generate hazardous secondary pollutants. The effective variable considered in the scenarios is the change in the type of fuel used in the thermoelectric power plants of Manaus, a metropolis located in the Amazon forest. The results show a marked decrease in the O3 level as a consequence of the fuel switching from diesel to natural gas mainly.

Recommendations to the Atmos. Chem. Phys.:

[Figure]

My recommendation to the journal ACP is the publication of the manuscript after a mandatory minor revision considering the following points:

Specific Comments:

* The authors could provide two additional explanatory paragraphs about the models "Regional Acid Deposition Model" Version 2 (RADM2) (Chang, 1991) and Model for Ozone and Related Tracers (MOZART-4) (Emmons et al., 2010).

* Would it to be interesting to indicate if the emission rates of the thermoelectric plants refer to an uninterrupted operation?

* The inclusion of a graph for the observed annual cycle (e.g., with box plots) of the pollutants considered for Manaus, to permit a in deepth discussion of the meteorological conditions which was related to the month period analysed.

* Characterizing the period of investigation from the meteorological point of view, in its synoptic and mesoscale aspects. Explaining the perturbations observed in the concentrations of pollutants observed (time series) in relation to the observed precipitation and possibility of wet deposition on the surface. In relation to the days without precipitation characterizing the mechanismof dry removal of the pollutants, qualitatively.

* Could the authors indicate that as a result of the wet removal process, the pollutants can be deposited/transferred on the surface, vegetation or indeed sent to the rivers. It could be indicated the superficial route when the meteorological conditions are of precipitation. It would be interesting to point out possible conservation routes rather than simply saying that the levels were decreasing. It could be explained this point better. Also the authors could also compare the concentration levels of sequential days in which the second day rains in Manaus. Or at least present a qualitative discussion.

* Presenting a better discussion on levels of background concentration associated with natural production by the forest and rivers.

Technical correction:

* Page 11 / line 255 Fix the orthografic flaw:"... The large diffe7rences ..."

---

## Referee Comment (RC1) · Anonymous Referee #1 · 13 Mar 2017

The manuscript 'Power plant fuel switching and air quality in a tropical forested environment' presented three sensitivity simulations to demonstrate that switching from fuel oil/diesel to natural gas as power energy can reduce maximum afternoon ozone In Manaus, Brazil. The idea, approach, and analysis are not novel, as the paper lacks sufficient new science to contribute what we know about ozone chemistry in a forested environment. Normally I would not recommend its publication in ACP. However I do see the potential interests of such a study for Manaus, a fast growing city surrounded by tropical forests. The shifting in power energy matrix is very exciting and is expected to have substantial impact on air quality, particularly in Amazonia region. Should it be published in ACP, a major revision is suggested to provide more science for ozone chemistry and/or anthropogenic emissions in tropical forested environment. A few suggestions but not limited as follow could really improve the quality of the paper:

[Figure]

– Validation of ozone and its precursors. We need to have at least some confidence on the model performance in this region. Given the paper was submitted to the GoAmazon2014/5 Special Issue, including some field campaign data for model evaluation would give the model credibility. A full evaluation of the model probably would be out of scope of the paper, but again, we need to at least know if the model does a fine job.

– There is some effort involved in development the emission inventory for this region, but again it is unclear to me how realistic the inventory would be. I would give the paper enough credit for just developing and presenting a realistic emission inventory for the region, given that widely used global inventories may do a terrible job here. This could be very useful for others doing research for this region too. According to the Table 4, basically NOx emissions are predominately contributed by power plants, with vehicles only contributing to 0.3%-3%. This is the fundamental to justify the ozone sensitivity simulations, and would need to be assessed.

– The discussion and analysis can be improved. One suggestion would be to demonstrate how ozone production efficiency changes among sensitivity simulations.

---

## Author Comment (AC1)

**Response to reviews**

Reviewer comments are in **bold**. Author responses are in plain text. Modifications to the manuscript are in *italics*. Page and line numbers in the responses correspond to those in the ACPD paper.

**Review #1**

**The manuscript 'Power plant fuel switching and air quality in a tropical forested environment' presented three sensitivity simulations to demonstrate that switching from fuel oil/diesel to natural gas as power energy can reduce maximum afternoon ozone In Manaus, Brazil.**

**1 - The idea, approach, and analysis are not novel, as the paper lacks sufficient new science to contribute what we know about ozone chemistry in a forested environment. Normally I would not recommend its publication in ACP.**

We thank the reviewer for this perspective. We are not aware of similar studies on a tropical forested region like Amazonas. We believe that the present study is important for the tropical context. Any additional literature that the reviewer is aware of would be appreciated by us.

**2 - However I do see the potential interests of such a study for Manaus, a fast growing city surrounded by tropical forests. The shifting in power energy matrix is very exciting and is expected to have substantial impact on air quality, particularly in Amazonia region.**

We thank the reviewer for this recognition.

**3 - Should it be published in ACP, a major revision is suggested to provide more science for ozone chemistry and/or anthropogenic emissions in tropical forested environment. A few suggestions but not limited as follow could really improve the quality of the paper:**

We thank the reviewer for this perspective. The reviewer has highlighted ozone chemistry and/or anthropogenic emissions as the focus of the manuscript, but the author's intent for what we wish to highlight and accomplish in this manuscript is actually different. Ozone and anthropogenic emissions were meant by us as a "means" but not an objective. In this context, the "objective" is to use the prime opportunity of a rapid switch in energy matrix to have a look at general effects on air quality in a tropical, forested environment, and in that regard we believe that this study is the first of its kind. By our intent, ozone and anthropogenic emissions inventories are incidental or a "means to an end" to this primary objective. In this light, we authors needed to clarify the manuscript. The introduction is revised with the following clear statement:

Section 1, Line 96:

*The study herein evaluates how a changing energy matrix in a tropical, forested environment affects urban pollutant concentrations. Ozone is chosen for detailed study because of the concern for human health and the susceptibility of its secondary production to factors at play in a forest environment. Manaus is chosen for study because of its location in the tropical forest, its size, and its shifting energy matrix. A large international experiment, Observations and Modeling of the Green Ocean Amazon (GoAmazon2014/5), was also carried out across two years in 2014 and 2015 in the Manaus region (Martin et al., 2016b), including aircraft flights (Martin et al., 2016a). A companion study by Rafee et al. (2017) compared simulated to measured pollutant concentrations for GoAmazon2014/5. The present study, which investigates how a shift in the energy matrix across a ten-year period affects regional air quality, provides interpretative context for the two-year experiment of GoAmazon2014/5.*

**4 - Validation of ozone and its precursors. We need to have at least some confidence on the model performance in this region. Given the paper was submitted to the GoAmazon2014/5 Special Issue, including some field campaign data for model evaluation would give the model credibility. A full evaluation of the model probably would be out of scope of the paper, but again, we need to at least know if the model does a fine job.**

We agree with the reviewer that although a detailed analysis is out of scope a comparison between model and observation is important. Figure S2 was added to the Supplement to compare GoAmazon2014/5 observations and the present study's simulated ozone. Material was also added to the main text.

Section 3, Line 227:

*As a check on the model output, a comparison between aircraft measurements of ozone concentrations and model predictions for Case B is presented in Figure S2*

**5 - There is some effort involved in development the emission inventory for this region, but again it is unclear to me how realistic the inventory would be. I would give the paper enough credit for just developing and presenting a realistic emission inventory for the region, given that widely used global inventories may do a terrible job here. This could be very useful for others doing research for this region too.**

We thank the reviewer for acknowledgment of the value of the study.

**6 - According to the Table 4, basically NOx emissions are predominately contributed by power plants, with vehicles only contributing to 0.3%-3%. This is**

**the fundamental to justify the ozone sensitivity simulations, and would need to be assessed.**

We thank the reviewer for this perspective. However, we think that extra sensitivity simulations are out of scope and does not attend to our main objective. We evaluated the change in energy matrix, specifically on fuel burned by power plants. For cases B and C the reductions of $NO_x$ emissions were of 60% and 89% in relation to Case A. Therefore, ozone production is sensitive to these changes in the simulations.
About the vehicle contribution in Table 4 (i.e., about 450 kg day$^{-1}$), the values are coherent to what has been reported in the literature proportionally to vehicle number and fleet age in Mexico City (Peritore, 1999;Molina and Molina, 2004). This fact emphasizes the importance of the paper because the main emission component (power plants) is going through a real fuel change in an unprecedented way in a tropical environment.

**7. The discussion and analysis can be improved. One suggestion would be to demonstrate how ozone production efficiency changes among sensitivity simulations.**

Although undoubtedly valuable, for the present study we believe the reviewer's suggestion is outside the scope of our objective. In this regard, please see response 6. The RADM2 chemical mechanism used in the simulations is developed and tested, and chamber experiments have been performed to evaluate the sensitivity and accuracy of chemical mechanism. Nevertheless, sensitivity simulations were performed previously by our group in other studies (Rafee et al., 2015;Rafee et al., 2017) changing the emissions precursors in 15% and 30%. The results demonstrated that the ozone concentration has a sensitivity to those variations as well $PM_{10}$ and $NO_x$. Ozone concentrations do not linearly changes as observed in general for $NO_x$ and $PM_{10}$ in response to changing emissions.

**References**

Martin, S., Artaxo, P., Machado, L., Manzi, A., Souza, R., Schumacher, C., Wang, J., Biscaro, T., Brito, J., and Calheiros, A.: The Green Ocean Amazon Experiment (GoAmazon2014/5) Observes Pollution Affecting Gases, Aerosols, Clouds, and Rainfall over the Rain Forest, Bulletin of the American Meteorological Society, 2016a.

Martin, S. T., Artaxo, P., Machado, L. A. T., Manzi, A. O., Souza, R. A. F., Schumacher, C., Wang, J., Andreae, M. O., Barbosa, H. M. J., Fan, J., Fisch, G., Goldstein, A. H., Guenther, A., Jimenez, J. L., Pöschl, U., Silva Dias, M. A., Smith, J. N., and Wendisch, M.: Introduction: Observations and Modeling of the Green Ocean Amazon (GoAmazon2014/5), Atmos. Chem. Phys., 16, 4785-4797, 10.5194/acp-16-4785-2016, 2016b.

Molina, M. J., and Molina, L. T.: Megacities and atmospheric pollution, Journal of the Air & Waste Management Association, 54, 644-680, 2004.

Peritore, N. P.: Third world environmentalism: case studies from the global South, University Press of Florida, 1999.

Rafee, S. A. A., Kawashima, A. B., de Morais, M. V. B., Urbina, V., Martins, L. D., and Martins, J. A.: Assessing the Impact of Using Different Land Cover Classification in Regional Modeling Studies for the Manaus Area, Brazil, Journal of Geoscience and Environment Protection, 3, 77, 2015.

Rafee, S. A. A., Kawashima, A. B., Almeida, D. S., Urbina, V., Morais, M. V. B., Souza, R. V. A., Oliveira, M. B. L., Souza, R. A. F., Medeiros, A. S. S., Freitas, E. D., Martins, D. L., and Martins, J.: Mobile and stationary sources of air pollutants in the Amazon rainforest: a numerical study with WRF-Chem model, submitted, Atmos. Chem. Phys, 2017.

---

## Author Comment (AC2)

**Response to Short Comment**

Reviewer comments are in **bold**. Author responses are in plain text. Modifications to the manuscript are in *italics*. Page and line numbers in the responses correspond to those in the ACPD paper.

**Short Comment #1**
**The manuscript brings an interesting contribution to the evaluation of the air pollution conditions associated with the production of thermoelectric energy in Manaus, Amazonas, Brazil. In particular, it presents modelling evidence based on an optimistic scenario of reducing the levels of atmospheric pollutants that participate in photochemical reactions to generate hazardous secondary pollutants. The effective variable considered in the scenarios is the change in the type of fuel used in the thermoelectric power plants of Manaus, a metropolis located in the Amazon forest. The results show a marked decrease in the O3 level as a consequence of the fuel switching from diesel to natural gas mainly. Recommendations to the Atmos. Chem. Phys.: My recommendation to the journal ACP is the publication of the manuscript after a mandatory minor revision considering the following points:**

The authors thank the reviewer for reading the manuscript and for the specific comments and the technical corrections. The final version of the manuscript takes into account the aspects pointed out. Responses to the specific points follow below.

**Specific Comments:**
**1 - The authors could provide two additional explanatory paragraphs about the models "Regional Acid Deposition Model" Version 2 (RADM2) (Chang, 1991) and Model for Ozone and Related Tracers (MOZART-4) (Emmons et al., 2010).**

We agree with the reviewer on this comment. The following sentence was added to the manuscript.

Section 2.1, Line 152.

*For Domain 2, the widely used Regional Acid Deposition Model Version 2 (RADM2) served as the chemical mechanism. It included 63 chemical species, 21 photolysis reactions, and 124 chemical reactions (Stockwell et al., 1990;Chang, 1991). Initial and boundary conditions for trace gases in Domain 2 were obtained from MOZART-4, an offline chemical transport model that has 85 chemical species, 12 aerosol compounds, 39 photolysis reactions and 157 gas-phase reactions (Emmons et al., 2010).*

**2 - Would it to be interesting to indicate if the emission rates of the thermoelectric plants refer to an uninterrupted operation?**

We agree with the reviewer on the importance of this detail, and the following sentence was inserted in the manuscript:

Section 2.3, Line 200:

*For the Manaus region, the power plants generate energy uninterruptedly at full load throughout the year, with contractual arrangements with industry to idle when residential demand increases.*

**3 - The inclusion of a graph for the observed annual cycle (e.g., with box plots) of the pollutants considered for Manaus, to permit a in depth discussion of the meteorological conditions which was related to the month period analysed.**

We thank the reviewer for this perspective. The authors are not aware of measured ozone annual cycle in Manaus urban zone. We do see the importance of characterize background ozone conditions in the wet season for the region, which was done in section 1. In this regard, please see response 7.

**4 - Characterizing the period of investigation from the meteorological point of view, in its synoptic and mesoscale aspects.**

The authors agree with the reviewer comment. The following sentence was added to the main text:

Section 2.1, Line 144:

*For this region, the climatological rainfall in February is 290 mm, which can be compared to a maximum of 335 mm in March and a minimum of 47 mm in August (Ramos et al., 2009). For February 2014, observed precipitation was 21.5% below the climatological value (Figure S1), as explained by the positioning of the Bolivian High to the west of its usual location (CPTEC-INPE, 2014).*

**5 - Explaining the perturbations observed in the concentrations of pollutants observed (time series) in relation to the observed precipitation and possibility of wet deposition on the surface. In relation to the days without precipitation characterizing the mechanism of dry removal of the pollutants, qualitatively.**

The authors agree with the reviewer comment. A new Figure S1 shows with observed daily rain amount, and a following sentence was added to the main text:

Section 3, Line 268:

*The observed daily rain amounts (Figure S1) show that the days having the highest ozone concentrations corresponded to days of low or no precipitation (<5 mm). Conversely, the days of highest precipitation (>20 mm) and cloudiness had nearly background ozone concentrations.*

**6 - Could the authors indicate that as a result of the wet removal process, the pollutants can be deposited/transferred on the surface, vegetation or indeed sent to the rivers. It could be indicated the superficial route when the meteorological conditions are of precipitation. It would be interesting to point out possible conservation routes rather than simply saying that the levels were decreasing. It could be explained this point better. Also the authors could also compare the concentration levels of sequential days in which the second day rains in Manaus. Or at least present a qualitative discussion.**

We thank the reviewer for that comment. Low ozone concentrations during the wet season are mostly linked to high cloudiness, leading to a lack of radiation and reduced rates of photochemical reactions and ozone production ($NO_2 + h\nu = NO + O$ and $O + O_2 + M = O_3 + M$). Ozone is a soluble gas, but the formation is dependent of radiation and the concentrations of VOC and NOx. In Manaus atmosphere due to high humidity and probably the large availability of OH, others chemical reactions should be important (e.g., $NO_2 + OH + M = HNO_3 + M$), decreasing the ozone production. The chemical mechanism used in the simulations considered these reactions. However, this subject was not addressed in this work.

**7 - Presenting a better discussion on levels of background concentration associated with natural production by the forest and rivers.**

We agree with the reviewer on this. A paragraph is already in the main text (Section 1, Line 75) discussing precursor emissions and ozone production over the forest. In response to the reviewer's suggestion and to further clarify the manuscript, the following additional sentence was added to the.

Section 1, Line 75:

[revised manuscript text omitted]

---

## Author Response (AR2)

**Response to Co-Editor coments**

Reviewer comments are in **bold**. Author responses are in plain text. Modifications to the manuscript are in *italics*. Page and line numbers in the responses correspond to those in the ACPD paper.

**1 - Reviewer 1, Point 4 - Figure S2 does not indicate if the model is actually doing a good job; please be more specific and discuss the validation**

We thank the reviewer for the perspective.

To respond both to this question as well as the next (see 2 below), aircraft data from GoAmazon2014/5 are used in the updated manuscript.

Figure S2(a) is included to show the comparison between aircraft and simulated data for the ozone precursor $NO_x$.

The discussion about Figure S2(b) has also been further developed.

Section 3, Line 248:

*A comparison between measurements of $NO_x$ and $O_3$ onboard the G-1 aircraft over and downwind of Manaus during GoAmazon2014/5 and simulation results is presented in Figure S2 for Case B. Figure S2a shows agreement between median and interquartile ranges of observed and simulated $NO_x$ concentrations. These concentrations above the natural regional background arise as primary pollutant in Manaus emissions. Likewise, simulated $O_3$ concentrations also show good agreement with aircraft data (Figure S2b). Ozone is a secondary pollutant, and the agreement supports the validity of the emission inventory of ozone precursors and the chemical mechanisms used in the simulation. Overall, the comparison shows that the simulations satisfactorily represent average regional afternoon concentrations of $NO_x$ and $O_3$*

**2 - Reviewer 1, Point 5 – This point has not been answered. Reviewer 2 has stressed this point: "However, my other concern, that how realistic is new emission inventory is, hasn't been addressed. Why not include a comparison for ozone precursors in Figure S2 to address this? For almost any environment, if NOx emission is reduced by 60% or 89% in the model, simulated ozone levels will be for sure reduced dramatically. So, it is important to know if the new developed emission inventory/scenario can really reflect the real world. Again, this is fundamental to justify this study."**

Figure S2(a) shows good agreement between the $NO_x$ concentration measured by the aircraft and the mean simulated values of the present study, demonstrating that the

simulation developed herein is able to represent observed concentrations of $NO_x$ and $O_3$ (Response 1).

A full evaluation of the emissions inventory is out of scope of the present study.

**3 - Reviewer 2, Point 3 - You have at least the ozone concentrations during GoAmazon? Please use them.**

We acknowledge the Co-Editor comment.

There are no surface ozone data available for Manaus urban region during the study period of the manuscript.

Airborne ozone data are available for several flights.

A comparison between observed aircraft data and simulated ozone concentrations is included in Figure S2. The discussion about that figure also is improved in the revised manuscript. In this regard, please see Response 1.

**4 - Reviewer 2, Point 4 - The meteorological background needs more information. The reference to the Bolivian High is thrown in without further explanation; indeed, it is associated with low rainfall in the Amazon but is not the explanation, or the cause. Also, the reviewer asked for a description of eventual mesoscale systems acting in the region during the time period.**

We thank the Co-Editor for the comment.

In order to address this point, the following paragraph was rewritten and improved:

Section 2.1, Line 143

*The climatology has differences between dry and wet seasons, with minimum values of monthly precipitation reached in August (47 mm) and maximum values found in March (335 mm) (Ramos et al., 2009). The month considered herein (February) has a climatological average of 290 mm. For February 2014, there was a deficit of 21.5% for meteorological stations in Manaus (Figure S1), with high precipitation above 20 mm on five days. The February deficit might correlate with a shifted position of the Bolivian high to the west of its normal position. This anticyclonic circulation at high atmosphere is associated with latent heat release during austral summer (Silva Dias et al., 1983;Jones and Horel, 1990). By comparison, the Intertropical Convergence Zone (ITCZ) was at its climatological position in February 2014, and exceptional events related to the South Atlantic Convergence Zone (SACZ) or other frontal systems in the region were absent (CPTEC-INPE, 2014).*

We thank the Co-Editor and reviewers for their time, input, and resulting improvements of the manuscript.

**References**

CPTEC-INPE: Meteorological Bulletin of the Center for Weather Forecasting and Climatic Studies (CPTEC) of the Brazilian National Institute of Space Research (INPE): http://climanalise.cptec.inpe.br/~rclimanl/boletim/index0214.shtml acessed 04 Apr 2017, 14:37., 2014.

Jones, C., and Horel, J. D.: A circulação da Alta da Bolívia e a atividade convectiva sobre a América do Sul, Revista brasileira de Meteorologia, 5, 379-387, 1990.

Ramos, A. M., dos Santos, L. A. R., and Fortes, L. T. G.: Normais climatológicas do Brasil, 1961-1990, 2009.

Silva Dias, P. L., Schubert, W. H., and DeMaria, M.: Large-scale response of the tropical atmosphere to transient convection, Journal of the Atmospheric Sciences, 40, 2689-2707, 1983.